# Risk assessment of temporary pacing for cardiac arrest after cardiopulmonary bypass-assisted cardiovascular surgery: A case-control study

Heng Wang[ID][1][◉]*, Li Shen[ID][1][◉], Qingwen Lin[2][◉], Heng Yu[3], Yu Zhang[1], Luzheng Zhang[1], Yujin Sun[1], Song Xue[1]*

**1** Department of Cardiovascular Surgery, Shanghai Jiao Tong University School of Medicine Affiliated Renji Hospital, Shanghai, China, **2** Dmir Lab, Guangdong University of Technology, Guangzhou, China, **3** Department of Clinical Laboratory, Shanghai Jiao Tong University School of Medicine Affiliated Renji Hospital, Shanghai, China

◉ These authors contributed equally to this work.
* nbwh113@163.com (HW); xuesong64@163.com (SX)

## Abstract

### Objective

Cardiac arrest happens in 0.7%-5.2% patients after cardiovascular surgery, and cases with asystole or severe bradycardia need timely temporary pacing. However, routine temporary pacing wire insertion in cardiopulmonary bypass (CPB)-assisted cardiovascular surgery has been questioned for its noteworthy complications. This study aimed to quantify the risk of temporary pacing for cardiac arrest after CPB-assisted cardiovascular surgery.

### Methods

2326 patients undergoing CPB-assisted cardiovascular surgery were enrolled. Age, sex, body mass index, preoperative rhythm, operation type, ablation, CPB pump, cardioplegia type and volume, hypothermia, circulation, CPB time, aortic clamping time were compared between patients having and not having temporary pacing according to the indications by multiple logistic regression (MLR). A scoring system was developed based on the β parameters of identified independent risk factors in MLR analyses. The score cutoff was determined by the negative likelihood ratio to exclude the need of temporary pacing.

### Results

108 patients (4.6%) had temporary pacing. Old age (per year) (P<0.001), preoperative atrial fibrillation (P<0.001), long CPB time (per minute) (P=0.017) contributed to the risk of cardiac arrest. Having mitral valve replacement (MVR) (P=0.033), double valve replacement (DVR), MVR+tricuspid valvuloplasty (TVP) (P=0.009), coronary

**Data availability statement:** Data cannot be shared publicly because of potential risks of having personal data privacy problems with combined personal information. Data are available from the Shanghai Jiao Tong University School of Medicine Affiliated Renji Hospital Institutional Ethics Committee (contact via guoyunyue@renji.com) for researchers who meet the criteria for access to confidential data. Requests for research data access can also be sent to the authors.

**Funding:** The accumulation of cases and data was supported in part by Shanghai Pudong New Area Health Commission Special Program for Clinical Research in the Health Industry [PW2010D-2, PW2015D-2, PW2021E-04]; Three-year Action Plan to Promote Clinical Skills and Clinical Innovation Capabilities in Municipal Hospitals, Shanghai Shenkang Hospital Development Center [SHDC2020CR6013]; Clinical Innovation and Training Funding of Shanghai Jiao Tong University School of Medicine Affiliated Renji Hospital [RJPY-DZX-005]. The funders had no role in study design, data collection and analysis, decision to publish, or preparation of the manuscript.

**Competing interests:** Potential competing interests are declared that Dr. Song Xue received financial supports from Shanghai Pudong Health Commission Special Program for Clinical Research in the Health Industry [PW2010D-2, PW2015D-2, PW2021E-04]; Three-year Action Plan to Promote Clinical Skills and Clinical Innovation Capabilities in Municipal Hospitals, Shanghai Shenkang Hospital Development Center [SHDC2020CR6013]; Clinical Innovation and Training Funding of Shanghai Jiao Tong University School of Medicine Affiliated Renji Hospital [RJPY-DZX-005]. Other authors declare that they have no conflict of interest. This does not alter our adherence to PLOS ONE policies on sharing data and materials

artery bypass grafting (CABG)+MVR (P = 0.0495) (versus CABG) were independent risk factors. The scoring system, score = *age (year)*/40 + *CPB time (min)*/350+ [*preoperative atrial fibrillation*]×1, can quantitatively assess the associated risk with an area under receiver of characteristic (ROC) curve (AUC) of 0.74 (95% confidential interval 0.69–0.79) (P < 0.001). The negative likelihood ratio was < 0.1 when score≤1.138. Therefore, the cutoff of excluding temporary pacing was set as ≤1, which achieved a 0% false negative rate in our cases.

## Conclusion

To minimize iatrogenic complications caused by unnecessary temporary pacing wire insertion, while ensuring patients with risks of asystole or severe bradycardia receive timely pacing, surgeons may identify cases with negligible risks of cardiac arrest through the scoring system.

---

## 1 Introduction

Cardiac arrest is a rare but severe complication after cardiovascular surgery [1]. In some cases characterized by asystole or profound bradycardia with a hemodynamic disturbance that is resistant or contraindicated to medication or other treatments, temporary pacing should be applied for regulating heart rhythm in resuscitation [2,3]. Although generally believed safe, routine temporary pacing wire insertion has been questioned due to its underestimated complications including tamponade, myocardial damage, and infection [4–6]. However, the evidence-based protocol recommended to start pacing within 1 minute if available once diagnosed with cardiac arrest [2]. Therefore, the potential risk factors should be investigated.

We hypothesized that perioperative characteristics including patient conditions, operation types, and operation parameters are related with postoperative cardiac arrest. This single center, case-control study aimed to assess the risk of temporary pacing after cardiopulmonary bypass (CPB)-assisted cardiovascular surgery on arrival to the cardiovascular surgery intensive care unit (CSICU).

## 2 Methods

### 2.1 Patients

The inclusion criteria were (1) undergoing successful CPB assisted cardiovascular surgery, (2) having preoperative and postoperative electrocardiogram results. The exclusion criteria were (1) necessary information was unidentifiable or lost, (2) having artificial cardiac pacing before operation, (3) with preoperative ventricular arrhythmia. We had patients in the medical record system of our department whose operations were done from January 1st, 2011 to July 31st, 2022. The records were collected during the perioperative period and administrated by 10 members (in which 4 members worked during the whole study) in our CPB team. The data were recorded and managed according to the clinical research projects, which ensured a minimal recall bias. Patients with and without medication resistant or contraindicated cardiac arrest

requiring temporary pacing were set as case group and control group, respectively. Patients in both case and control groups underwent operations assisted by the same CPB team, which ensured consistent CPB procedures and outcome estimation. This study was approved by Ethical Committee of our institution (LY2024–209-B) (27/08/2024) with waiver of informed consent due to retrospective nature of the study. The research reported in this paper adhered to Helsinki Declaration (2013).

## 2.2 Data collection

Whether having cardiac arrest requiring temporary pacing, defined as asystole or severe bradycardia of ventricular rate <60beats/min along with cardiac index < $2.2L\cdot min^{-1}\cdot m^{-2}$ and is resistant or contraindicated to epinephrine, dopamine administration, was recorded in each patient on arrival to CSICU by the Swan-Ganz catheter and electrocardiogram monitoring, respectively, after operation as binary outcomes. (1) Patient conditions including sex, age, height, weight, and preoperative rhythm (determined by electrocardiogram before thoracotomy), (2) operation types, whether having surgical ablation, and (3) CPB parameters including CPB pump type, cardioplegia type, cardioplegia volume, hypothermia strategy (the core body temperature was lowered to (1) mild, 32°C-35°C; (2) moderate, 22°C-32°C; (3) deep, < 22°C [7]), circulation, CBP time, aortic clamping time were recorded and accessed from the recording system on 28/08/2024 for the research purpose.

## 2.3 Multiple logistic regression (MLR)

MLRs were done with main effects and the intercept, and the models were assessed using area under receiver of characteristic (ROC) curve (AUC), Hosmer-Lemeshow test, log-likelihood ratio test. Male sex, occlusive pump, crystal cardioplegia were set as reference levels in the MLR for their lowest incidences in the corresponding characteristics; sinus rhythm, no ablation, mild hypothermia, normal circulation were set as reference levels for their relatively physiological statuses in the corresponding characteristics; coronary artery bypass grafting (CABG) was set as the reference level in the operation type for its external, non-valvular feature of procedure that less affects the conduction system.

## 2.4 Sensitivity analysis

To evaluate the robustness of risk factor(s), 3 sensitivity analyses were performed after the initial MLR. (1) Using different models: continuous variables were replaced by their square-rooted, squared, and cubic forms and reanalyzed in MLRs. (2) Using data without outliers: outliers in continuous variables were identified by the ROUT method (Q = 1%) or iterative Grubbs' method (alpha = 0.05), and cases without outliers were reanalyzed by MLRs. (3) Trying data binning: the age was grouped into young (0 to 65 years, ref.) and old (>65 years); the BMI ($kg\cdot m^{-2}$) was grouped into underweight (BMI < 18.5), normal (18.5 ≤ BMI < 24.0, ref.), overweight (24 ≤ BMI < 28), obesity (≥28); the CPB and aortic clamping times (min) were grouped into short (<90, < 60, ref.), medium (90 to 120, 60 to 90), long (>120, > 90) according to the distribution and convention; the cardioplegia volume (ml) was grouped into small (<350, ref.), medium (350 to 1500), large (>1500) according to the distribution, and they were reanalyzed in an MLR (the reference levels were chosen for their relatively physiological statuses).

## 2.5 Scoring system

The scoring system was developed based on the independent risk factors found in the MLR screening. The operation type was not included in the scoring system for its uncertain, complicated compositions. To simplify the scoring process, characteristics were excluded by stepwise elimination. Briefly, to predict the binary outcome of whether requiring temporary pacing for cardiac arrest, (1) sex, age (years), ablation, BMI ($kg\cdot m^{-2}$), pump type, CPB time (min), aortic clamping time (min), cardioplegia type, cardioplegia volume (ml), hypothermia strategy, circulation, preoperative rhythm were included in the first MLR, and the characteristic with the largest P value was excluded in the second screening (if the discrete

variable has ≥ 2 levels, the P value for exclusion was determined by the smallest one); (2) in the second MLR, the characteristic with the largest P value among the rest of characteristics was excluded in the third screening; (3) the screening was continuously processed until all variables in a MLR were of statistical significance, and these variables were used in the scoring system. The MLR screening result provides β parameters for the logarithmic odds of the temporary pacing probability, and the scales of scoring points of independent risk factors were determined by their relative β parameter values. The cutoff of binary prediction was determined by the Youden's index (finding the cutoff that maximizes the value sensitivity+specificity-1).

### 2.6 Subgroup analysis

Since the operation type was not included in characteristic screening, the scoring system was applied to groups of cases undergoing different operation types including mitral valve replacement (MVR)+tricuspid valvuloplasty (TVP), CABG+MVR, double valve replacement (DVR) (replacing both mitral valve and aortic valve), MVR, mitral valvuloplasty (MVP), DVR+TVP, aortic valve replacement (AVR), CABG, and the rest of operation types (Other).

### 2.7 Likelihood ratios

The positive likelihood ratios (sensitivity/(1-specificity)) and the negative likelihood ratios ((1-sensitivity)/specificity) were calculated with different cutoffs of scores. A cutoff with a positive likelihood ratio >10 or a negative likelihood ratio <0.1 was considered acceptable for confirming or excluding the risk of cardiac arrest, respectively.

### 2.8 Statistical analysis

Data were analyzed using GraphPad Prism 9.5. Data of continuous variables are presented as "median (lower and upper quartiles)". In the demographics and operational information, the normality of all continuous variables was checked by D'Agostino & Pearson test; all continuous variables between groups were compared by Mann-Whitney test; data of discrete variables are presented as "frequency (percentage in row)", and were analyzed by chi-square test (>2 categories) or Fisher's exact test (2 categories); the odds ratio (OR) and its 95% confidence interval (95%CI) were determined by single logistic regression. The 95%CI of incidence of temporary pacing was estimated using Wilson-Brown method. When applicable, data were analyzed two-sided. *P<0.05, **P<0.01, ***P<0.001 are considered of statistical significance.

## 3 Results

### 3.1 Patients

As is shown in Fig 1, during the research period, our institution had 5184 patients undergoing successful CPB-assisted cardiovascular surgery. 2446 patients with preoperative and postoperative rhythms were included. The potential participates were not included because the records were collected by specific members in the CPB team during their working times. 109 patients with incomplete information, 9 patients with preoperative pacing, 2 patients with preoperative ventricular arrhythmia (1 had ventricular tachycardia; 1 had degree III atrioventricular (AV) block) were excluded, and 2326 patients were enrolled in which 108 patients (4.6%) had the cardiac arrest requiring temporary pacing.

### 3.2 Risk factors

As is shown in Table 1, patients who needed temporary pacing had older age, longer CPB and aortic clamping times; preoperative atrial fibrillation and undergoing surgical ablation had higher odds with statistical significance; operation types having top 9 number of cases were isolated in the analysis (other types were mostly combined procedures), and case and control groups were found to have different compositions of operation types; a larger cardioplegia volume of marginal statistical significance was found in the case group. In the initial MLR (Fig 2, S1 Table), the model had an AUC of 0.76

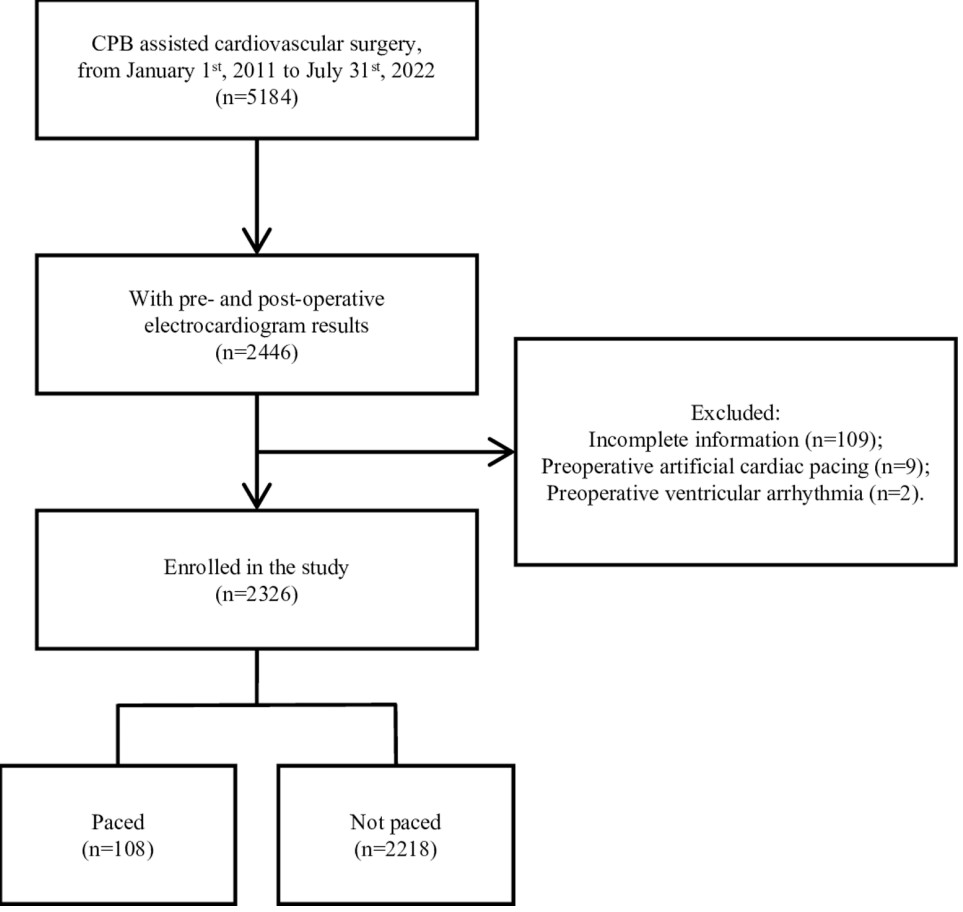

**Fig 1. The flow chart of enrollment.** The records were retrieved from successful operations. All paced patients met the indication of temporary pacing, i.e. asystole or severe bradycardia of ventricular rate <60beats/min along with a cardiac index <2.2L·min$^{-1}$·m$^{-2}$, of temporary pacing. Abbreviation: CPB, cardiopulmonary bypass.

(95%CI 0.72 to 0.80), and it passed the Hosmer-Lemeshow test (P = 0.957) and log-likelihood ratio test (P < 0.001); old age (per year), long CPB time (per min) were found to independently contribute to temporary pacing. Preoperative atrial fibrillation was found as an independent risk factor. Compared with CABG, patients undergoing MVR, DVR, MVR + TVP, CABG+MVR had higher risks of temporary pacing. In the sensitivity analyses, (1) when continuous variables were replaced by their cubic values, MVR did not show a higher risk of temporary pacing compared with CABG (S3 Table); (2) when continuous variables were replaced by their squared or cubic values, CPB time did not contribute to the temporary pacing risk (S2 Table and S3 Table); when continuous variables were grouped, CPB time over 120min was an independent risk of temporary pacing, while CPB time between 90min and 120min did not increase the risk, compared with CPB time shorter than 90min (S7 Table); (4) other risk factors found in the initial MLR were not changed in sensitivity analyses.

### 3.3 Risk assessment

After the stepwise elimination, preoperative rhythm, age (years), CPB time (min) remained to be the characteristics included in the scoring system, and the MLR result indicated that their β parameters were 1.5, 0.037, and 0.0042, respectively (Table 2). To make the scoring process easy, the scores of sinus rhythm and atrial fibrillation in preoperative rhythm

**Table 1. Demographics and operational information*.**

| Characteristic | Temporary pacing | | P value | Odds ratio (95%CI) |
|---|---|---|---|---|
| | Yes (n = 108) | No (n = 2218) | | |
| **Sex** | | | | |
| Male | 51 (4.0) | 1219 (96.0) | P = 0.137 | Ref. |
| Female | 57 (5.4) | 999 (94.6) | | 1.36 (0.93-2.01) |
| **Age (year)** | 64 (56-70) | 58 (46-66) | P < 0.001 | 1.04 (1.02-1.05) |
| **BMI (kg·m⁻²)** | 22.0 (19.7-25.2) | 22.4 (20.0-24.8) | P = 0.610 | 0.99 (0.94-1.04) |
| **Preoperative rhythm** | | | | |
| Sinus rhythm | 58 (3.0) | 1872 (97.0) | P < 0.001 | Ref. |
| Atrial fibrillation | 50 (12.6) | 346 (87.4) | | 4.66 (3.13-6.92) |
| **Operation** | | | | |
| CABG | 2 (1.1) | 185 (98.9) | P < 0.001 | Ref. |
| MVR | 19 (5.8) | 310 (94.2) | | 5.67 (1.62-35.83) |
| AVR | 7 (2.9) | 231 (97.1) | | 2.80 (0.67-18.96) |
| DVR | 14 (6.7) | 194 (93.3) | | 6.68 (1.83-42.87) |
| MVR+TVP | 20 (10.4) | 173 (89.6) | | 10.69 (3.06-67.59) |
| MVP | 5 (4.5) | 107 (95.5) | | 4.32 (0.91-30.53) |
| CABG+MVR | 6 (8.0) | 69 (92.0) | | 8.04 (1.80-55.78) |
| DVR+TVP | 3 (4.3) | 66 (95.7) | | 4.21 (0.68-32.45) |
| ASD closure | 1 (2.0) | 48 (98.0) | | 1.93 (0.09-20.52) |
| Other | 31 (3.6) | 835 (96.4) | | 3.43 (1.03-21.32) |
| **Ablation** | | | | |
| Yes | 26 (11.0) | 210 (89.0) | P < 0.001 | Ref. |
| No | 82 (3.9) | 2008 (96.1) | | 3.03 (1.88-4.76) |
| **Pump** | | | | |
| Occlusive | 106 (4.6) | 2182 (95.4) | P = 0.696 | Ref. |
| Centrifugal | 2 (5.3) | 36 (94.7) | | 1.14 (0.18-3.81) |
| **Cardioplegia type** | | | | |
| Crystal | 7 (4.3) | 154 (95.7) | P > 0.999 | Ref. |
| Cold blood | 101 (4.7) | 2064 (95.3) | | 1.08 (0.53-2.59) |
| **Cardioplegia volume (ml)** | 600 (400-1600) | 500 (350-1500) | P = 0.055 | 1.00 (0.99-1.00) |
| **Hypothermia** | | | | |
| Mild | 84 (4.9) | 1642 (95.1) | P = 0.634 | Ref. |
| Moderate | 18 (4.2) | 409 (95.8) | | 0.86 (0.50-1.41) |
| Deep | 6 (3.5) | 167 (96.5) | | 0.70 (0.27-1.50) |
| **Circulation** | | | | |
| Normal | 103 (4.8) | 2054 (95.2) | P = 0.345 | Ref. |
| Arrested or low-flow | 5 (3.0) | 164 (97.0) | | 0.61 (0.21-1.37) |
| **CPB time (min)** | 115.5 (91.0-148.8) | 105.0 (79.0-141.0) | P = 0.008 | 1.00 (1.00-1.01) |
| **Aortic clamping time (min)** | 75.0 (52.5-92.0) | 65.0 (45.0-90.0) | P = 0.030 | 1.00 (1.00-1.01) |

* Abbreviation: ASD, atrial septal defect; AVR, aortic valve replacement; BMI, body mass index; CABG, coronary artery bypass grafting; CI, confidence interval; CPB, cardiopulmonary bypass; DVR, double valve replacement; MVP, mitral valvuloplasty; MVR, mitral valve replacement; TVP, tricuspid valvuloplasty.

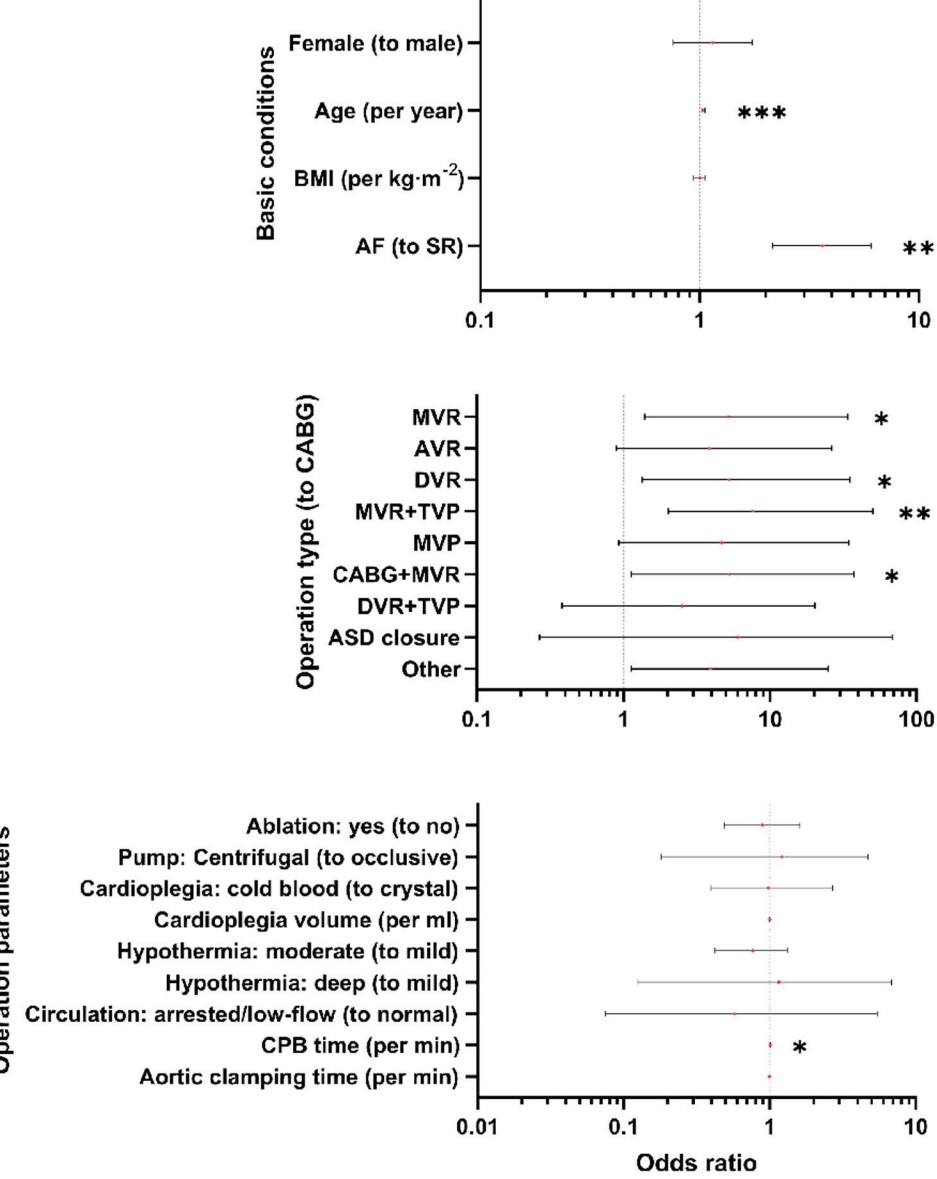

**Fig 2. Risk factors in the initial multiple logistic regression.** Age (per year) (OR 1.04, 95%CI 1.02 to 1.06) (P<0.001), CPB time (per min) (OR 1.01, 95%CI 1.00 to 1.01) (P=0.017) added to the risk of temporary pacing. Compared with SR, preoperative AF was an independent risk factor of temporary pacing (OR 3.64, 95%CI 2.16 to 6.06) (P<0.001). Compared with CABG, MVR (OR 5.22, 95%CI 1.40 to 34.06) (P=0.033), DVR (OR 5.28, 95%CI 1.34 to 35.16) (P=0.036), MVR+TVP (OR 7.67, 95%CI 2.02 to 50.41) (P=0.009), CABG+MVR (OR 5.28, 95%CI 1.13 to 37.47) (P=0.0495) increased the risk of temporary pacing. Abbreviations: AF, atrial fibrillation; ASD, atrial septal defect; AVR, aortic valve replacement; BMI, body mass index; CABG, coronary artery bypass grafting; CI, confidence interval; CPB, cardiopulmonary bypass; DVR, double valve replacement; MVP, mitral valvuloplasty; MVR, mitral valve replacement; OR, odds ratio; SR, sinus rhythm; TVP, tricuspid valvuloplasty.

were set as 0 and 1, respectively, and the scores of age (per year) and CPB time (per min) were scaled to be $0.024 \approx 1/40$ and $0.0028 \approx 1/350$. The score of temporary pacing risk can be calculated by the formula score = *age (year)*/40 + *CPB time (min)*/350 + [*preoperative atrial fibrillation*]×1. The scoring system achieved an AUC of 0.74 (95%CI 0.69 to 0.79) (P<0.001) (Fig 3A), and scores of case and control groups were 2.38 (1.94–2.85) and 1.84 (1.50–2.18) (P<0.001) (Fig 3B), respectively. Sensitivity and specificity were 59.26% and 80.16%, respectively, at the cutoff of score=2.282

**Table 2. The scoring system\*.**

| Characteristic | Parameter (point) | Odds ratio (95%CI) | P value |
|---|---|---|---|
| **Preoperative rhythm** | | | |
| Sinus rhythm | Ref. (0) | Ref. | NA |
| Atrial fibrillation | 1.503 (1) | 4.496 (3.008-6.702) | P<0.001 |
| **Age (per year)** | 0.03672 (1/40) | 1.037 (1.021-1.055) | P<0.001 |
| **CPB time (per min)** | 0.004231 (1/350) | 1.004 (1.001-1.007) | P=0.006 |
| **Intercept** | -6.150 (NA) | 0.0021 (0.0006-0.0064) | P<0.001 |

\* Abbreviation: CI, confidence interval; CPB, cardiopulmonary bypass; NA, not applicable.

determined by the Youden's index. The incidence of temporary pacing increased with the score (Fig 3C). When the score was below 2, the incidence remained below 5% (Fig 3C). The subgroup analyses indicated that the scoring system performed well in MVR + TVP, CABG+MVR, MVR, AVR, CABG, and other groups, while no statistical significance was found in DVR, MVP, DVR + TVP groups (Fig 4), which suggests that its performances in different operation types can be varied.

### 3.4 Likelihood ratios

With an interval of 0.002, cutoff values ≤1.138 or ≥3.474 corresponded to negative likelihood ratios <0.1 and positive likelihood ratios >10 (Fig 5A). To ensure an effective and simple identification of patients not requiring temporary pacing, we chose a score cutoff of 1, i.e., the prediction of temporary pacing requirement is negative when score≤1, and otherwise the prediction is positive. With this strategy, no paced case was predicted to be negative and 161 cases were excluded from temporary pacing (Fig 5B). This model had 100% positive recall and 0% false negative rate, while the positive precision was 4.99% (Fig 5C).

## 4 Discussion

### 4.1 The dilemma of routine temporary pacing wire insertion

The incidence of cardiac arrest after cardiovascular surgery was reported to be 0.7%-5.2% and half of the postoperative cardiac arrest happened within 3 hours [2]. In pediatric patients after cardiovascular surgery, cardiac arrest can increase the mortality from 2.8% to 49.4% [8]. Therefore, although the incidence was low, cardiovascular surgeons tend to routinely place the temporary epicardial pacing wire in high-risk operations, e.g. the CPB-assisted procedures, to promptly start temporary pacing. However, there is a 1.74% incidence of complications including wire retention, arrhythmia, delayed discharge, and cardiac tamponade [9]. Considering a high proportion of patients that do not develop cardiac arrest, these iatrogenic complications can be a noteworthy problem. In pediatric patients undergoing congenital cardiac surgery, severe postoperative arrhythmia can be rare [10]. Some surgeons well predicted the temporary pacing requirement and encouraged determining epicardial pacing wire insertion after evaluation in these pediatric patients [11,12]. Routine temporary pacing wire insertion has also been believed overused in CABG for the low incidence of severe postoperative arrhythmia [13–16]. Other cases can be more complicated. However, even though previous researches found several factors increasing the risk of cardiac arrest, there is no reliable quantitative method that excludes the need of temporary pacing, and decisions made by the current evidence can be venturesome.

### 4.2 The advantage of scoring and decisions based on the likelihood ratio analysis

Epinephrine had no longer been recommended by the end of our study period, which extended the indication of temporary pacing [2]. Our study focused on temporary pacing for cardiac arrest after CPB-assisted cardiovascular surgery on arrival to CSICU. Epicardial pacing wires were routinely inserted in our center during the study period and temporary

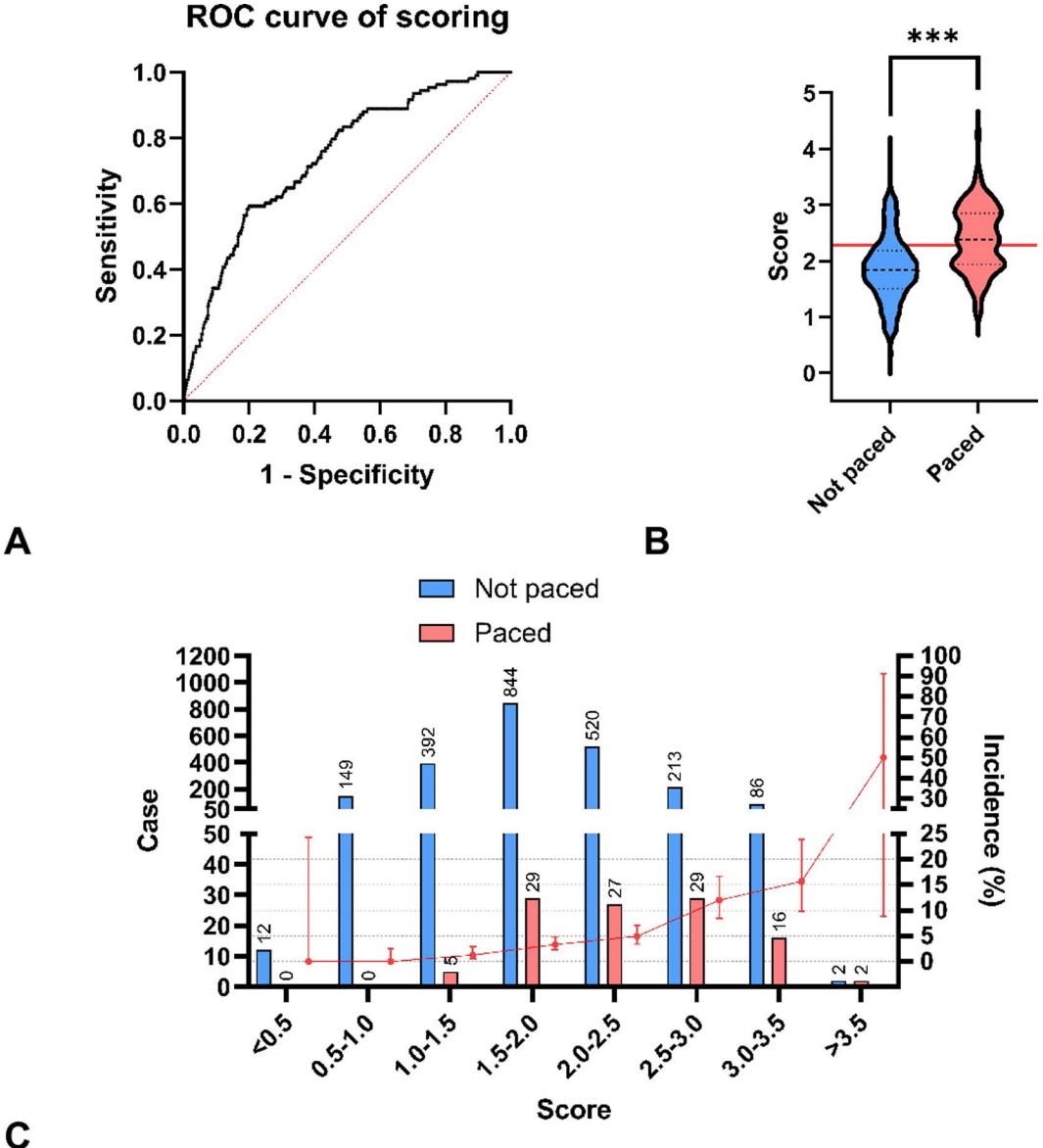

**Fig 3. The scoring system based on chosen independent risk factors.** (A) The ROC curve. (B) Scores of patients paced and not paced according to the indication where the red line marks the cutoff of 2.282 determined by Youden's index (P<0.001 tested by Mann-Whitney test). (C) Case distribution and incidence of temporary pacing versus score levels. Abbreviation: ROC, receiver of characteristic.

pacing was applied to about 4.6% patients for medication resistant or contraindicated cardiac arrest. Consistent with a previous study, old age, long CPB time, preoperative atrial fibrillation found before thoracotomy, some non-CABG operations were independent risk factors of cardiac arrest among our chosen characteristics [17]. Other potential causal factors, e.g. reduced ventricular ejection fraction, high serum lactate level, pulmonary hypertension, New York Heart Association class III-IV, some medications, may also increase its risk [17–19]. A scoring system will be quite helpful to quantify the temporary pacing requirement, which can provide reliable evaluation for decision making. In this study, we established a scoring system, i.e., score = *age (year)*/40 + *CPB time (min)*/350+ [*preoperative atrial fibrillation*]×1, for the cardiac arrest risk assessment using β parameters of characteristics screened from the stepwise elimination in the MLR result. We did

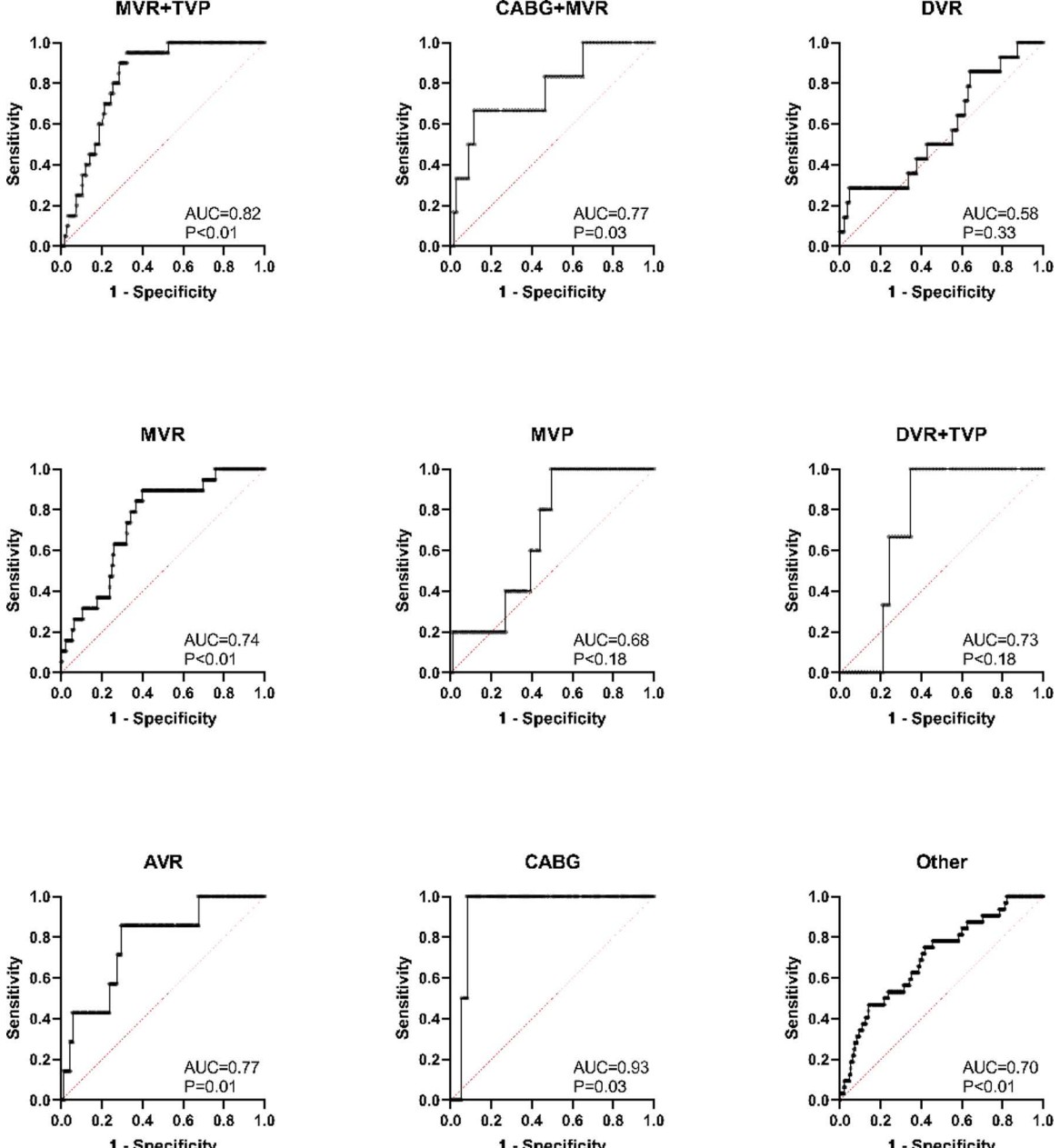

**Fig 4. The subgroup analyses.** The scoring system was applied to patients having different procedures. The ROC curve and the AUC with P value were calculated in each subgroup. Abbreviations: AUC, area under curve; AVR, aortic valve replacement; CABG, coronary artery bypass grafting; DVR, double valve replacement; MVP, mitral valvuloplasty; MVR, mitral valve replacement; ROC, receiver of characteristic; TVP, tricuspid valvuloplasty.

not include the operation type in the scoring system for several reasons. Firstly, the composition of operation types often varies across different centers; secondly, it is highly susceptible to the influence of the surgeon's skill; thirdly, incorporating this factor would increase the complexity of the scoring system, potentially hindering its clinical applicability. The mortality of cardiac arrest after cardiovascular surgery is high, and the exclusion of temporary pacing wire insertion should be carefully assessed to avoid severe events. Therefore, the binary prediction cannot be performed according to the Youden's

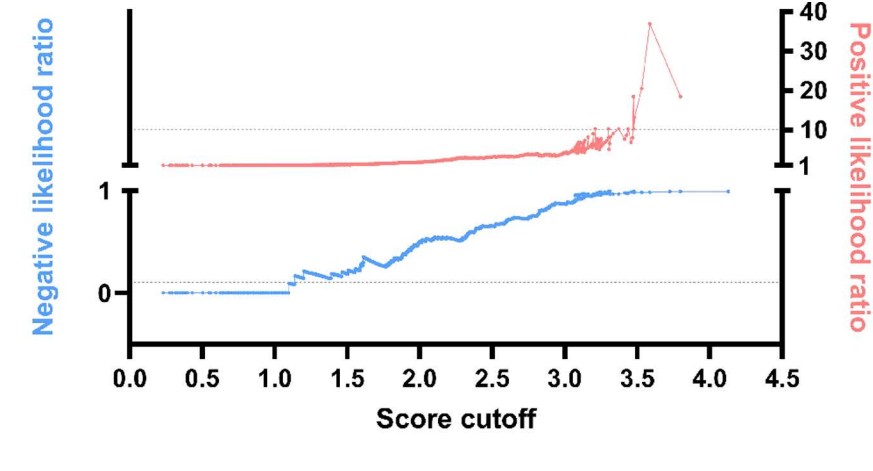

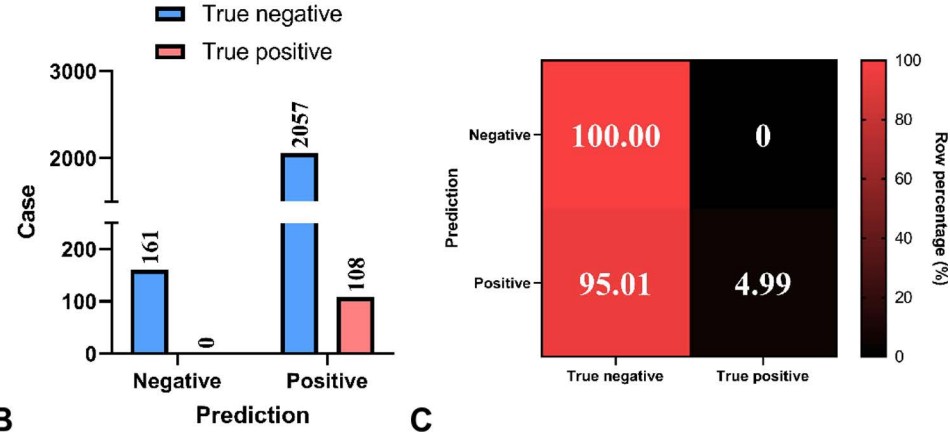

**Fig 5. Likelihood ratios and the exclusion cutoff performance.** (A) The likelihood ratio of scoring system where the dash lines mark the negative likelihood of 0.1 and the positive likelihood of 10. (B) The case distribution with the score cutoff of 1 in the prediction. (C) Percentages of true negative or positive cases in each prediction in (B).

index under this specific circumstance. Instead, the negative likelihood ratio is a good index to exclude a diagnosis [20]. We selected a cutoff of ≤1 as the exclusion range, which corresponded to a negative likelihood ratio <0.1 and achieved a 0% false negative rate. This indicates that for a patient younger than 40 years old without preoperative atrial fibrillation, the maximum acceptable CPB time without requiring temporary pacing wire insertion can be determined using the values provided in S8 Table. The score can also help surgeons or patients to make their decisions when it is > 1.

### 4.3 The underlying mechanisms

Valvular surgery is acknowledged to be a risk factor of AV block and can increase the risk of temporary pacing. From an anatomy perspective, valve-related procedures often carry a risk of AV block due to its proximity of the AV conduction axis, and surgical operations can cause iatrogenic injuries [21]. Technical advancements are needed to prevent such injuries. In contrary, CABG only has operations on the large branches of coronary arteries on the surface of the heart, which hardly causes damages to the conduction system in the deep layer, less requiring pacing compared with the valvular surgery. The risk factors of AV block after CABG includes old age, CPB factors and medication [22]. The future studies may group operation types into several categories according to these features or develop specialized evaluations for high risk

procedures. Atrial fibrillation was identified as a key risk factor of cardiac arrest in this study, and the potential mechanisms are (1) the re-entrant rhythm from atria can disturb the ventricular rhythm, causing asystole or ventricular fibrillation, and (2) there are shared genetics, comorbidities, habitats that increase risks of both atrial fibrillation and postoperative cardiac arrest [23]. Additionally, the influence of anti-atrial fibrillation medication should be considered as a confounding factor. Concomitant surgical ablation for atrial fibrillation can improve the late survival in patients undergoing cardiovascular surgery, while it may also cause right atrial lesion that needs pacing [24,25]. Interestingly, after adjusting, we found that preoperative atrial fibrillation rather than ablation increased the risk of postoperative temporary pacing. However, other studies suggest that surgical ablation may increase the risk of permanent pacing, which needs further verification [26,27]. The CPB time was also identified as a risk factor of temporary pacing. Except for the long-lasting non-physiological status under extracorporeal circulation, a long CPB time may result from other factors such as the procedure complexity and the surgeon's skill proficiency.

## 4.4 Clinical implementation

The prediction of postoperative cardiac arrest can fall into an accuracy paradox [28]. A model that classifies all cases into cardiac arrest negative group can simply achieve a high accuracy, because the incidence of cardiac arrest is very low. This is extremely harmful when surgeons neglect the risk of postoperative cardiac arrest based on small sample studies, considering the increased mortality that is avoidable. On the other hand, many efforts have been made to minimize the complications caused by routine wire insertion. Applying biodegradable materials is an active field, which can reduce the complication happening during wire removal [29–31]. Another strategy is to insert the wire under the guidance of echocardiography instead of routine wire insertion, which enables instant pacing initiation [32]. Recently, the millimetre-scale bioresorbable optoelectronic systems have been introduced, which may allow for injectable pacemaker deployment, ensuring rapid pacing in critical cases [33]. These technical advancements can optimize the clinical decisions (Fig 6), and we believe that evidence-based evaluation is always necessary under new circumstances. The ideal evaluation of pacing need should be simple, effective, clinically appropriate, and incorporating local epidemiological data into electronic health records to enable automated decision support would significantly enhance clinical utility.

## 4.5 Limitations

There are limitations in this study.

(1) Patients with the need of temporary pacing were found to have a higher risk of mortality [34], while the cardiac arrest and other outcomes in hospital and after discharge were not investigated in this study. Although cardiac arrest events mostly happen in the early period after operation and are the main reason of temporary pacing, future studies should focus on the pacing requirement before wire removal along with other indications of temporary pacing.

(2) Although the scoring system achieved a 0% false negative rate with the exclusion cutoff of ≤1, there was a large false positive rate which should be further optimized through including other risk factors or using other models.

(3) Differences in surgical techniques, disease spectrum, and other factors may influence the results, as the clinical outcomes of patients undergoing cardiovascular surgery vary significantly due to the complexity of practices, including in-hospital management, anesthesia, and CPB.

(4) This is a single-center study, and the conclusions should be validated using external datasets. To avoid selection bias from the small positive case sample size, we did not separate a validation group. However, testing the scoring system within the same dataset may overestimate the model performance. Future studies on machine learning models for the universal predicting purpose should ensure a sufficient sample size needed for complete model evaluation.

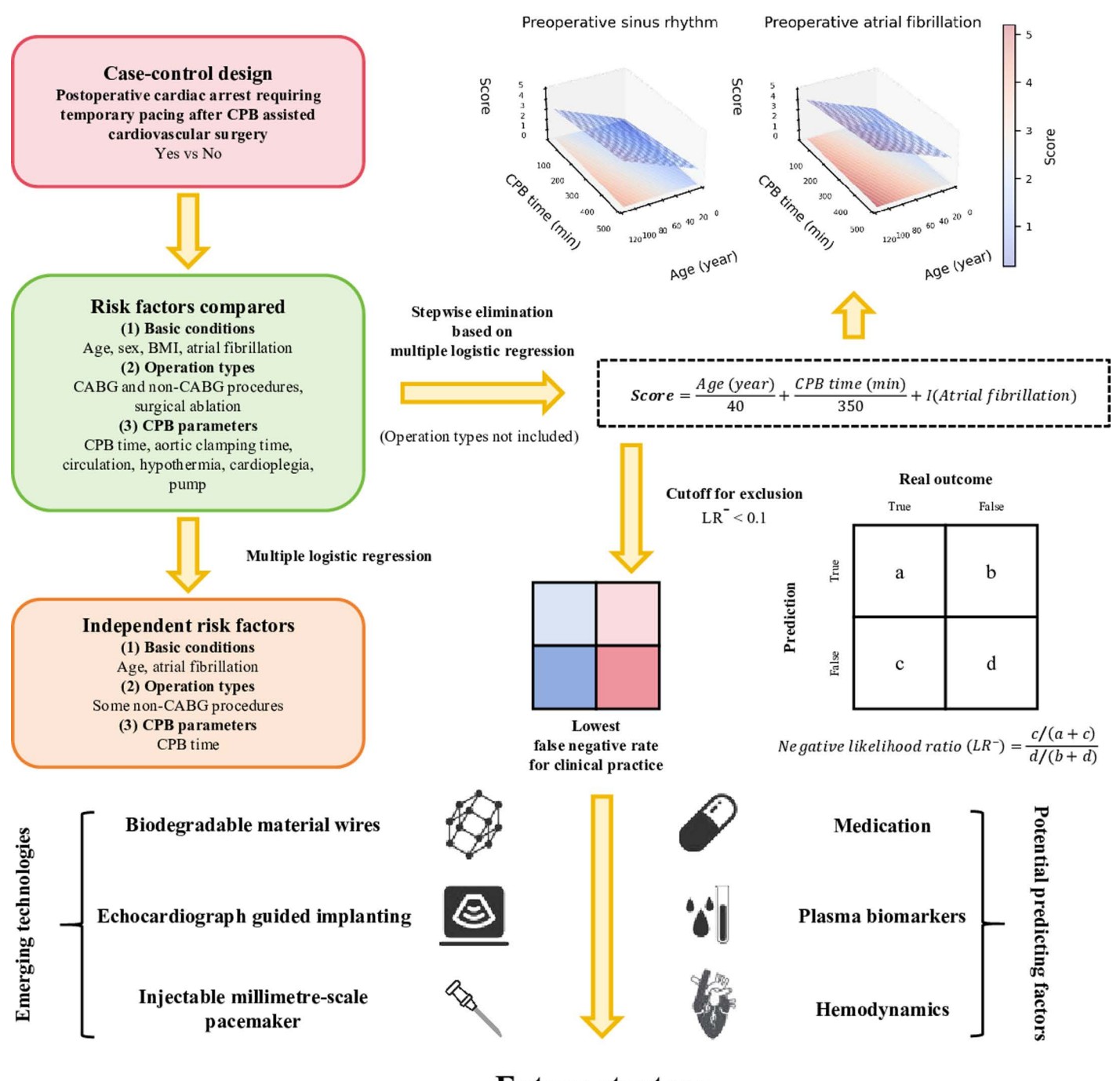

**Fig 6. Schematics of the scoring system development and prospects.** The function I in the score formula denotes an indicator function whose value is 1 when having preoperative atrial fibrillation; 0 when not having preoperative atrial fibrillation. The weights of age and CPB time were determined by the β parameters in the multiple logistic regression results. Abbreviations: BMI, body mass index; CABG, coronary artery bypass grafting; CPB, cardiopulmonary bypass.

(5) This is a retrospective study, which cannot avoid biases of selection (the data collection was done by specific members in their working times, and the case enrollment may be influenced by personal factors), information (including missing data bias caused by the update of recording systems, ascertainment bias in diagnoses, and personal factors), confounding (factors like calcium channel blocker using, perioperative echocardiography results, hemodynamics, comorbidities were not investigated in MLR analyses) that reduce the model generalizability in small sample subgroups. The development of more applicable quantitative model should consider tailored prospective cohort studies in multiple centers investigating consensus factors for clear causal inference. Randomized controlled trials for new pacing technologies may provide novel insights of this clinical issue and can minimize the selection bias.

## 5 Conclusion

Preoperative arrhythmia before thoracotomy, old age, long CPB time were risk factors of cardiac arrest after CPB-assisted cardiovascular surgery. The scoring system, i.e., score = *age (year)*/40 + *CPB time (min)*/350+ [*preoperative atrial fibrillation*]×1, can quantify this risk. Surgeons can use this system to exclude the need for temporary pacing wire insertion in patients with a score ≤1, indicating a low risk of cardiac arrest following cardiovascular surgery.

## Supporting information

**S1 Table. The initial multiple logistic regression.**
(DOCX)

**S2 Table. The multiple logistic regression with squared continuous variables.**
(DOCX)

**S3 Table. The multiple logistic regression with cubic continuous variables.**
(DOCX)

**S4 Table. The multiple logistic regression with square-rooted continuous variables.**
(DOCX)

**S5 Table. The multiple logistic regression without outliers identified by the ROUT method.**
(DOCX)

**S6 Table. The multiple logistic regression without outliers identified by the iterative Grubbs' method.**
(DOCX)

**S7 Table. The multiple logistic regression with binning continuous variables.**
(DOCX)

**S8 Table. The longest CPB time with limited temporary pacing risk in different ages.**
(DOCX)

## Acknowledgments

We appreciate the CPB team of Department of Cardiovascular Surgery, Shanghai Jiao Tong University School of Medicine Affiliated Renji Hospital for their data recording and management.

## Author contributions

**Conceptualization:** Song Xue.

**Data curation:** Li Shen.

**Formal analysis:** Heng Wang.

**Funding acquisition:** Song Xue.

**Investigation:** Yu Zhang.

**Methodology:** Qingwen Lin.

**Project administration:** Song Xue.

**Resources:** Song Xue.

**Visualization:** Heng Yu.

**Writing – original draft:** Heng Wang, Luzheng Zhang.

**Writing – review & editing:** Li Shen, Yujin Sun.

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
