## [Decision Letter · Decision Letter 0]

3 Mar 2025

PONE-D-24-58174Risk assessment of temporary pacing for cardiac arrest after cardiopulmonary bypass assisted cardiovascular surgery: a case-control studyPLOS ONE

Dear Dr. Wang,

Thank you for submitting your manuscript to PLOS ONE. After careful consideration, we feel that it has merit but does not fully meet PLOS ONE’s publication criteria as it currently stands. Therefore, we invite you to submit a revised version of the manuscript that addresses the points raised during the review process.

We look forward to receiving your revised manuscript.

Kind regards,

Eyüp Serhat Çalık

Academic Editor

PLOS ONE

“The accumulation of cases and data was supported in part by Shanghai Pudong New Area Health Commission Special Program for Clinical Research in the Health Industry [PW2010D-2, PW2015D-2, PW2021E-04]; Three-year Action Plan to Promote Clinical Skills and Clinical Innovation Capabilities in Municipal Hospitals, Shanghai Shenkang Hospital Development Center [SHDC2020CR6013]; Clinical Innovation and Training Funding of Shanghai Jiao Tong University School of Medicine Affiliated Renji Hospital [RJPY-DZX-005].”

“Potential competing interests are declared that Dr. Song Xue received financial supports from Shanghai Pudong Health Commission Special Program for Clinical Research in the Health Industry [PW2010D-2, PW2015D-2, PW2021E-04]; Three-year Action Plan to Promote Clinical Skills and Clinical Innovation Capabilities in Municipal Hospitals, Shanghai Shenkang Hospital Development Center [SHDC2020CR6013]; Clinical Innovation and Training Funding of Shanghai Jiao Tong University School of Medicine Affiliated Renji Hospital [RJPY-DZX-005]. Other authors declare that they have no conflict of interest.”

4. We note that there is identifying data in the Supporting Information file <Spreadsheet.1.xlsx>. Due to the inclusion of these potentially identifying data, we have removed this file from your file inventory. Prior to sharing human research participant data, authors should consult with an ethics committee to ensure data are shared in accordance with participant consent and all applicable local laws.

-Location data

Please remove or anonymize all personal information, ensure that the data shared are in accordance with participant consent, and re-upload a fully anonymized data set. Please note that spreadsheet columns with personal information must be removed and not hidden as all hidden columns will appear in the published file.

Additional Editor Comments:

I would like to thank the esteemed authors for presenting their work on this important topic for cardiac surgery. The manuscript was reviewed by 5 valuable referees, and their comments are below. We look forward to your point-by-point responses to the suggestions and your re-uploading of your manuscript after appropriate revisions.

Reviewers' comments:

Reviewer's Responses to Questions

**Comments to the Author**

1. Is the manuscript technically sound, and do the data support the conclusions?

Reviewer #1: Yes

Reviewer #2: Yes

Reviewer #3: Yes

Reviewer #4: Yes

Reviewer #5: Partly

2. Has the statistical analysis been performed appropriately and rigorously? 

Reviewer #1: No

Reviewer #2: Yes

Reviewer #3: Yes

Reviewer #4: N/A

Reviewer #5: No

3. Have the authors made all data underlying the findings in their manuscript fully available?

Reviewer #1: No

Reviewer #2: Yes

Reviewer #3: Yes

Reviewer #4: Yes

Reviewer #5: Yes

4. Is the manuscript presented in an intelligible fashion and written in standard English?

Reviewer #1: Yes

Reviewer #2: Yes

Reviewer #3: Yes

Reviewer #4: Yes

Reviewer #5: No

5. Review Comments to the Author

Reviewer #1: The abstract does not provide sufficient background to contextualize the importance of studying temporary pacing.

Suggestion: Add 1-2 sentences explaining the clinical implications of temporary pacing and its relevance to postoperative outcomes.

The results section is data-heavy, with numerous ORs and CIs presented in a dense format that might overwhelm readers.

While the abstract mentions the scoring system, it does not explain how the scoring system was developed or validated.

Lack of Specificity in the Conclusion:

The conclusion reiterates findings but does not address the scoring system’s potential for clinical implementation or its limitations.

Ambiguity in Results Interpretation:

Terms like "long CPB time" are vague and lack specific thresholds, which might limit reproducibility.

No Mention of Limitations:

The abstract does not acknowledge any study limitations, which could affect the interpretation of findings.

Terminology: Replace "Odd ratio" with "Odds ratio" for accuracy.

Scoring System AUC: While the ROC curve AUC of 0.7402 is acceptable, its clinical significance is not addressed. How well does the scoring system perform in different patient populations?

Preoperative Atrial Fibrillation: This is identified as a major risk factor. The abstract could mention its prevalence in the study cohort for context.

Reviewer #2: The study makes an important contribution, highlighting key risk factors for the need for temporary pacing upon CPB-assisted cardiovascular surgery and by developing a scoring system that could be clinically useful. However, the study could benefit from a deeper exploration of the mechanisms behind the identified risk factors. To enhance the clinical application and the robustness of the study, the following areas should be addressed:

1. This study provides clear exclusion criteria, there is a potential risk of collection bias, given that the data was collected by specific team members during working hours. The authors should clarify if there are inherent biases like selection bias, recall bias, and missing data issues, which could have affected the comprehensiveness of their findings.

2. While the authors report an AUC of 0.7400 for their scoring system, external validation is critical. In this case, where no external validation exists, caution must be emphasized, as validation in an independent cohort would test the robustness and generalizability of the scoring system. For example, the clinical effectiveness of this model needs to be evaluated prospectively in other hospitals or surgical settings. Furthermore, while the scoring system’s sensitivity (59.26%) and specificity (80.16%) are reported, it would be helpful to provide additional details on the performance of the model at different score cutoffs. How does the model perform in different patient subgroups (e.g., those with comorbidities, and specific valve surgeries)?

3. The authors highlight that non-CABG operations (e.g., MVR, DVR) are independent risk factors for temporary pacing. However, the paper does not discuss how the complexity or specific technical aspects of these procedures may increase the likelihood of pacing. For instance, does mitral valve replacement inherently cause more myocardial stress or post-operative arrhythmias? Further discussion could provide a more mechanistic understanding of these findings.

4. The study focuses on the need for temporary pacing, including other postoperative outcomes such as mortality, long-term arrhythmias, and recovery times. These additional outcomes would provide a more comprehensive assessment of the clinical implications of temporary pacing. The authors didn’t mention if temporary pacing leads to a need for permanent pacing in some patients. Including this information would be valuable, as it would indicate whether the need for temporary pacing might signal a higher risk for long-term pacing requirements.

Reviewer #3: Major Comments:

Study Design and Patient Selection:

The retrospective design is appropriate given the clinical context; however, additional clarification on the selection criteria (and potential selection biases) would strengthen the manuscript.

The exclusion of patients with incomplete information or preoperative pacing is clearly justified, but the manuscript should discuss how these exclusions might affect generalizability.

Statistical Analysis and Model Development:

The logistic regression analysis appears robust, and the identification of independent risk factors is supported by appropriate statistical tests.

The development of the scoring system is innovative; however, external validation in an independent cohort is necessary to confirm its clinical utility.

The manuscript would benefit from a more detailed discussion of the model’s calibration (beyond the Hosmer-Lemeshow test) and discrimination metrics.

Clinical Implications:

The authors effectively argue that temporary pacing is an important intervention with non-negligible risks, and that predicting its necessity could optimize patient management.

It would be useful to elaborate on how the scoring system might be integrated into clinical practice and whether it might reduce unnecessary temporary pacing wire insertions.

Minor Comments:

Some sections of the Methods and Results could be more concise; consider streamlining the text to focus on the most critical details.

The description of the risk factors (e.g., CPB time, preoperative rhythm) is clear, but providing a table that summarizes baseline characteristics between the pacing and non-pacing groups would enhance clarity.

Figures and supplementary materials (such as the ROC curve and distribution of scores) are helpful; ensure they are of high resolution and fully referenced in the text.

There are a few typographical errors and minor formatting issues that should be corrected prior to publication.

Reviewer #4: Review of the Manuscript: Risk Assessment of Temporary Pacing for Cardiac Arrest After Cardiopulmonary Bypass-Assisted Cardiovascular Surgery: A Case-Control Study

The manuscript presents a case-control study on the risk assessment of temporary pacing for cardiac arrest following cardiopulmonary bypass (CPB)-assisted cardiovascular surgery. The study addresses an important clinical question, offering a scoring system to evaluate risk factors. However, there are areas that require revision, including the clarity of the methodology, statistical justification, and overall presentation of results. Below are specific comments regarding strengths and areas for improvement.

Strengths:

1. Clinical Relevance: The study addresses a significant clinical concern, providing a risk assessment for temporary pacing after cardiac surgery.

2. Large Sample Size: The study includes 2,326 patients, which strengthens the statistical power.

3. Clear Identification of Risk Factors: Independent risk factors such as age, preoperative atrial fibrillation, and prolonged CPB time are well-identified and explained.

4. Development of a Scoring System: The creation of a risk stratification model adds practical value for clinical decision-making.

Major Concerns:

1. Abstract Clarity and Precision

• The abstract should explicitly state the statistical methods used to determine independent risk factors and scoring system development.

• Clarify the meaning of “certain indications” in the methods section.

• The conclusion should emphasize the clinical applicability of the scoring system.

2. Methodology and Statistical Analysis

• The description of how patients were categorized into case and control groups lacks clarity. The inclusion and exclusion criteria need refinement, particularly regarding preoperative ventricular arrhythmia.

• The statistical justification for using specific cut-offs in logistic regression models should be provided.

• The rationale for choosing coronary artery bypass grafting (CABG) as the reference category in operation type comparisons should be explained.

• Multiple logistic regression: The criteria for stepwise elimination should be described in greater detail.

• The scoring system parameters need validation against an external dataset. If not feasible, a discussion on its limitations should be included.

3. Presentation of Results

• Tables 1 and 2 should be reformatted for clarity. Consider summarizing key findings in the text.

• Figure 2D requires additional explanation to clarify how likelihood ratios were determined.

• Sensitivity and specificity should be interpreted in a clinical context—what constitutes an acceptable threshold?

4. Discussion and Clinical Implications

• The study should compare findings with existing literature on temporary pacing after CPB.

• Potential confounding factors such as perioperative medications, patient comorbidities, and intraoperative hemodynamics should be acknowledged.

• The limitations section should expand on how surgical techniques and institutional practices may affect generalizability.

5. Language and Formatting

• Several grammatical errors and awkward phrasings are present. A professional language review is recommended.

• Consistency in terminology: The manuscript alternates between “temporary pacing” and “temporary pacemaker use” without clear distinction.

• Reference formatting should follow journal guidelines.

Minor Comments:

1. Define abbreviations upon first use.

2. Ensure figures are high resolution and appropriately labeled.

3. The ethical approval statement should be moved to the methods section.

This study presents valuable findings on temporary pacing after CPB-assisted cardiovascular surgery. However, revisions are necessary to enhance methodological clarity, statistical justification, and the clinical interpretation of results. Addressing these concerns will significantly strengthen the manuscript's impact and readability.

Reviewer #5: Abstract:

Spell out the numbers two thousand three hundred twenty-six and one hundred eight as they appear at the start of the sentence.

Rephrase: A scoring formula was developed, achieving an area under the receiver operating characteristic (ROC) curve (AUC) of 0.7402 (95% CI: 0.6939–0.7865, P < 0.0001).

Revise: A scoring system incorporating age, preoperative rhythm, and CPB time can quantitatively assess the associated risk.

Methods:

Since this is a case-control study, explicitly state the inclusion and exclusion criteria for both cases and controls.

Justify why this study is not classified as a retrospective cohort study.

Revise the statistical analysis section: Specify where the D’Agostino & Pearson test, Mann-Whitney test, chi-square test or Fisher’s exact test, and Baptista-Pike method were applied in the results.

Results:

Spell out the numbers two thousand four hundred forty-six and one hundred nine as they appear at the start of the sentence.

Report p-values to three decimal places, percentages to one decimal place, and other values to two decimal places.

Table 1:

-This table should report results from simple logistic regression.

-Replace "statistics" with p-values.

-Reclassify some variables where necessary.

-Ensure the reference category is reported first (e.g., "male" followed by "female").

Table 3: Clearly state the variable selection method used.

Revise:

-The incidence of temporary pacing increased with the score. When the score was below 2, the incidence remained below 5% (Fig. 2C).

-With an interval of 0.002, cutoff values ≤1.138 or ≥3.474 corresponded to negative likelihood ratios <0.1 and positive likelihood ratios >10 (Fig. 2D).

Discussion:

Expand this section as it is currently too brief.

Revise the use of the term "incidence" to ensure accuracy.

Discuss why the type of operation was not included in the scoring system.

Address the exclusion of several key factors and how this impacts the scoring system.

References:

Additional references should be included where relevant.

6. PLOS authors have the option to publish the peer review history of their article (what does this mean? ). If published, this will include your full peer review and any attached files.

**Do you want your identity to be public for this peer review?** For information about this choice, including consent withdrawal, please see our Privacy Policy .

Reviewer #1: **Yes: ** Aram Baram

Reviewer #2: **Yes: ** Victory Ashonibare

Reviewer #3: No

Reviewer #4: **Yes: ** Juliana Aggrey

Reviewer #5: No

---

## [Author Response · Author response to Decision Letter 1]

10 Mar 2025

Responses to the reviewers

(1)Reviewer #1

Q1: The abstract does not provide sufficient background to contextualize the importance of studying temporary pacing. Suggestion: Add 1-2 sentences explaining the clinical implications of temporary pacing and its relevance to postoperative outcomes.

A1: We added the background that explains why we need to carry out this study in the Abstract-Objective part. The clinical context is also further introduced in the Discussion 4.1 part. Briefly, the key problem is that (1) the high risk (49.4%) of mortality caused by postoperative cardiac arrest should be timely rescued by temporary pacing although the incidence was low (0.7%-5.2%), but (2) routine pacing wire insertion can cause noteworthy (1.74%) iatrogenic complications, considering the large number of patients that do not develop a cardiac arrest. Our work aimed to identify as many patients that do not develop a cardiac arrest to reduce the use of pacing wires.

Q2: The results section is data-heavy, with numerous ORs and CIs presented in a dense format that might overwhelm readers.

A2: We adjusted data presenting in Table 1: (1) P values were mostly rounded to 3 decimal places; (2) original data were rounded to up to 2 decimal place. We also replaced the Table 2 with forest plots in Fig 2, which makes the results easy to be read.

Q3: While the abstract mentions the scoring system, it does not explain how the scoring system was developed or validated.

A3: We wrote separated sections, Method 2.5 and 2.7, to introduce how the scoring system was developed. The scoring system was validated using the same cases and its performance of excluding patients that do not develop cardiac arrest achieved 0% false negative rate, which meets the clinical need.

Q4: The conclusion reiterates findings but does not address the scoring system’s potential for clinical implementation or its limitations.

A4: We rewrote the conclusion in Abstract and the main text. The clinical implementation was clarified as the scoring system was established to exclude the patients with low risks of cardiac arrest. The limitations were placed in the Discussion part 4.4.

Q5: Terms like "long CPB time" are vague and lack specific thresholds, which might limit reproducibility.

A5: To make our results clear, we provided the S8 Table. It guides readers to find the longest CPB time that does not significantly increase the risk of cardiac arrest in different ages.

Q6: The abstract does not acknowledge any study limitations, which could affect the interpretation of findings.

A6: We agree with your opinion that the lack of limitations in Abstract may cause a misinterpretation of results, while with all necessary information presented, we reached the 300-word limit of Abstract. We ensured that the limitations are fully discussed in the Discussion 4.4 part.

Q7: Replace "Odd ratio" with "Odds ratio" for accuracy.

A7: Thank you for pointing out this spelling mistake. It has been corrected in the text and figures.

Q8: Scoring System AUC. While the ROC curve AUC of 0.7402 is acceptable, its clinical significance is not addressed. How well does the scoring system perform in different patient populations?

A8: As is guided by the STROBE guidance, we added subgroup analyses in the Result 4.4 part to analyze the system’s performances in different patient populations.

Q9: Preoperative Atrial Fibrillation: This is identified as a major risk factor. The abstract could mention its prevalence in the study cohort for context.

A9: The prevalence of atrial fibrillation was 396 (17.02% in all cases, and 50 in paced cases). It is necessary to write the prevalence of characteristic when it has limited absolute number, while here the atrial fibrillation prevalence was not small (low) in both case and control groups, and we believe that the data did not cause a misinterpretation. Considering limited words in Abstract, please understand that we cannot add the data there.

(2)Reviewer #2

Q1: This study provides clear exclusion criteria, there is a potential risk of collection bias, given that the data was collected by specific team members during working hours. The authors should clarify if there are inherent biases like selection bias, recall bias, and missing data issues, which could have affected the comprehensiveness of their findings.

A1: We agree that there are biases in our study. Since the data collection details are required according to the STROBE guidance, we wrote the members and details of data collection, and ensured that the standard data collection strategies planned by the supporting projects were obeyed, which well minimized the selection bias. In our opinion, the most “solid” bias may originate from the retrospective nature of our study and we fully acknowledged the possible biases, including the ones you mentioned here, in the limitation part.

Q2: While the authors report an AUC of 0.7400 for their scoring system, external validation is critical. In this case, where no external validation exists, caution must be emphasized, as validation in an independent cohort would test the robustness and generalizability of the scoring system. For example, the clinical effectiveness of this model needs to be evaluated prospectively in other hospitals or surgical settings. Furthermore, while the scoring system’s sensitivity (59.26%) and specificity (80.16%) are reported, it would be helpful to provide additional details on the performance of the model at different score cutoffs. How does the model perform in different patient subgroups (e.g., those with comorbidities, and specific valve surgeries)?

A2: We agree that an external set of cohort for validation is crucial, especially in some machine learning models. The difficulty is that our study topic was rarely studied currently and we cannot find a similar research with individual data. An alternative way is to separate a group of cases as the external set, however, the major problem is that the positive (cases who were paced) sample size is small (108 cases) as some subgroups only have less than 5 positive cases, which can cause significant selection bias and be questioned by other researchers. Therefore, we chose to test our scoring system in the same cases, which has been commonly used in medical studies.

In our study, we investigated 2 cutoffs. The first cutoff, as is mentioned here (Sen 59.26%, Spe 80.16%), was determined by the Youden’s index, and this method approaches the balanced, largest Sen and Spe values, however, it is not suitable in the clinical use, because if a patient who finally develops a cardiac arrest but he or she did not have a wire inserted, there can be severe events. The second cutoff was determined by the negative likelihood ratio (please see Method 2.7 and Result 3.4), and this cutoff identifies the patients who can be excluded from wire insertion, which can wrongly identify patients who are cardiac arrest-negative as cardiac arrest-positive but never identify those who are positive as negative. The second cutoff is more helpful here, because it helps patients and surgeons to efficiently avoid a part of unnecessary wire insertion.

We did not have enough comorbidity data for analysis, but we performed subgroup analyses in different subgroups of procedures. We agree that the subgroup analyses are necessary, because having the valve surgery was identified as a risk factor but it was not included in the scoring system.

Q3: The authors highlight that non-CABG operations (e.g., MVR, DVR) are independent risk factors for temporary pacing. However, the paper does not discuss how the complexity or specific technical aspects of these procedures may increase the likelihood of pacing. For instance, does mitral valve replacement inherently cause more myocardial stress or post-operative arrhythmias? Further discussion could provide a more mechanistic understanding of these findings.

A3: Thanks for your reminding. We did not deeply think about the reason why non-CABG operations can increase the risk of temporary pacing. Here we inserted the reference [21] and briefly introduced the most acceptable explanation currently available, i.e. the valvular surgery procedures can cause injuries in atrioventricular conduction axis because the replacing/repairing area is close to this axis. Additionally, we also discussed other risk factors in the Discussion part.

Q4: The study focuses on the need for temporary pacing, including other postoperative outcomes such as mortality, long-term arrhythmias, and recovery times. These additional outcomes would provide a more comprehensive assessment of the clinical implications of temporary pacing. The authors didn’t mention if temporary pacing leads to a need for permanent pacing in some patients. Including this information would be valuable, as it would indicate whether the need for temporary pacing might signal a higher risk for long-term pacing requirements.

A4: Due to limited literature we found and data we have, we cannot draw a conclusion that permanent pacing is positively correlated with temporary pacing, but studies on single temporary or permanent pacing have found common risk factors. Here we inserted a report ([27]) published on BJCVS, and it found that the in hospital mortality was 10% higher in temporary paced patients compared with those who were not paced. We put it into the Limitations part and admit that this is a shortage of our study.

(3)Reviewer #3

Q1: The retrospective design is appropriate given the clinical context; however, additional clarification on the selection criteria (and potential selection biases) would strengthen the manuscript. The exclusion of patients with incomplete information or preoperative pacing is clearly justified, but the manuscript should discuss how these exclusions might affect generalizability.

A1: In the Limitation part, we wrote that selection bias can influence the results through personal factors. As for the exclusion criteria, your advice remind us of a key problem that our recording system was once updated in 2018, which may influence the data recording process and this potential bias was also added to the Limitation part. Some patients were excluded for preoperative ventricular arrhythmia and artificial pacing, while this proportion of patients is quite small and the patients’ basic conditions have a definite influence on the outcome, thus we believe that their exclusion does not affect the model’s generalizability.

Q2: The logistic regression analysis appears robust, and the identification of independent risk factors is supported by appropriate statistical tests. The development of the scoring system is innovative; however, external validation in an independent cohort is necessary to confirm its clinical utility. The manuscript would benefit from a more detailed discussion of the model’s calibration (beyond the Hosmer-Lemeshow test) and discrimination metrics.

A2: Since Prime 9.5 only provides limited tools, we added 2 additional parts for the model’s calibration according to the STROBE guideline. In Method part 2.4+Result part 3.2, we performed the sensitive analyses to assess the robustness of the independent risk factors we found. In Method part 2.6+Result 3.3, we applied the scoring system to different operation type subgroups to assess the system’s generalizability in different patients.

Q3: The authors effectively argue that temporary pacing is an important intervention with non-negligible risks, and that predicting its necessity could optimize patient management. It would be useful to elaborate on how the scoring system might be integrated into clinical practice and whether it might reduce unnecessary temporary pacing wire insertions.

A3: In the Discussion 4.1 part, we wrote the problem of current clinical strategy: (1) the high risk (49.4%) of mortality caused by postoperative cardiac arrest should be timely rescued by temporary pacing although the incidence was low (0.7%-5.2%), but (2) routine pacing wire insertion can cause noteworthy (1.74%) iatrogenic complications, considering the large number of patients that do not develop a cardiac arrest. In the Discussion 4.2 part, we wrote the reason why the quantitative scoring system along with the cutoff determined by negative likelihood ratio is useful and better than some “traditional” binary outcome prediction models, as we only exclude patients with a very low risk of cardiac arrest. This strategy is quite practical in clinical decisions.

Q4: Some sections of the Methods and Results could be more concise; consider streamlining the text to focus on the most critical details.

A4: In this revision, we separated multiple logistic regression, sensitivity analysis, scoring system establishing, subgroup analysis, likelihood ratio calculation into different parts. The results include 4 parts: general information of the patients, risk factor verification, the scoring system’s performance, and its use in the clinical context with a cutoff determined by negative likelihood ratio. This organization can better illustrate the contents.

Q5: The description of the risk factors (e.g., CPB time, preoperative rhythm) is clear, but providing a table that summarizes baseline characteristics between the pacing and non-pacing groups would enhance clarity.

A5: Please find the baseline information in Table.1. Recently, we read that some researchers advised not comparing demographics information in Table.1, but we still made statistical analyses comparing the non-pacing and pacing groups. We did not draw conclusions from this table since there can be a Simpson's paradox in non-RCT cohorts.

Q6: Figures and supplementary materials (such as the ROC curve and distribution of scores) are helpful; ensure they are of high resolution and fully referenced in the text. There are a few typographical errors and minor formatting issues that should be corrected prior to publication.

A6: Thanks for your advice. Before this submission, we checked the fig quality using the tool provided by the journal. We carefully checked the spelling and grammar in the revised manuscript.

(4)Reviewer #4

Q1: Abstract Clarity and Precision

• The abstract should explicitly state the statistical methods used to determine independent risk factors and scoring system development.

• Clarify the meaning of “certain indications” in the methods section.

• The conclusion should emphasize the clinical applicability of the scoring system.

A1: We reorganized the Abstract. The methods of statistical analyses (specifically, multiple logistic regression), scoring system establishing and the study’s clinical applicability was added to the Method part and the Conclusion part, respectively. We wrote a background in the Objective part that clarifies the indications (asystole or severe bradycardia) of pacing.

Q2: Methodology and Statistical Analysis

• The description of how patients were categorized into case and control groups lacks clarity. The inclusion and exclusion criteria need refinement, particularly regarding preoperative ventricular arrhythmia.

A2-1: We verily recorded these processes, and although they are simpler than prospective studies, no additional criterion or standard was applied. Since the study design is case-control, the categorizing process may be mostly influenced by the information bias; the inclusion and exclusion processes may be influenced by the selection bias, and we added these possible problems in the Limitation part. The 2 preoperative ventricular arrhythmia cases were ventricular tachycardia and degree III atrioventricular block, respectively, and we added this information in the Result 4.1 part.

• The statistical justification for using specific cut-offs in logistic regression models should be provided.

A2-2: We described the Youden’s index calculation that determined the cutoff (2.282) with Sen 59.26%, Spe 80.16% in the Method 2.5 part. Another cutoff (≤1) was determined by the negative likelihood ratio that can exclude the need of temporary pacing, which is described in Method 2.7 part.

• The rationale for choosing coronary artery bypass grafting (CABG) as the reference category in operation type comparisons should be explained.

A2-3: We added the reason “for its external, non-valvular feature of procedure that less affects t

---

## [Decision Letter · Decision Letter 1]

8 Apr 2025

PONE-D-24-58174R1Risk assessment of temporary pacing for cardiac arrest after cardiopulmonary bypass-assisted cardiovascular surgery: a case-control studyPLOS ONE

Dear Dr. Wang,

Thank you for submitting your manuscript to PLOS ONE. After careful consideration, we feel that it has merit but does not fully meet PLOS ONE’s publication criteria as it currently stands. Therefore, we invite you to submit a revised version of the manuscript that addresses the points raised during the review process.

Please submit your revised manuscript by May 23 2025 11:59PM. If you will need more time than this to complete your revisions, please reply to this message or contact the journal office at plosone@plos.org . Please include the following items when submitting your revised manuscript:

We look forward to receiving your revised manuscript.

Kind regards,

Eyüp Serhat Çalık

Academic Editor

PLOS ONE

Journal Requirements:

Additional Editor Comments:

I would like to thank the authors for their point-by-point responses to the reviewers' suggestions and for their appropriate revisions. We recommend a small revision for additional suggestions. Good luck.

Reviewers' comments:

Reviewer's Responses to Questions

**Comments to the Author**

1. If the authors have adequately addressed your comments raised in a previous round of review and you feel that this manuscript is now acceptable for publication, you may indicate that here to bypass the “Comments to the Author” section, enter your conflict of interest statement in the “Confidential to Editor” section, and submit your "Accept" recommendation.

Reviewer #2: All comments have been addressed

Reviewer #4: All comments have been addressed

Reviewer #5: (No Response)

2. Is the manuscript technically sound, and do the data support the conclusions?

Reviewer #2: Yes

Reviewer #4: Yes

Reviewer #5: Partly

3. Has the statistical analysis been performed appropriately and rigorously? 

Reviewer #2: Yes

Reviewer #4: Yes

Reviewer #5: No

4. Have the authors made all data underlying the findings in their manuscript fully available?

Reviewer #2: Yes

Reviewer #4: Yes

Reviewer #5: Yes

5. Is the manuscript presented in an intelligible fashion and written in standard English?

Reviewer #2: Yes

Reviewer #4: Yes

Reviewer #5: Yes

6. Review Comments to the Author

Reviewer #2: (No Response)

Reviewer #4: General Overview

This study aims to assess the risk of temporary pacing in patients experiencing cardiac arrest after CPB-assisted cardiovascular surgery. The authors employ multiple logistic regression (MLR) to identify independent risk factors and develop a scoring system for clinical application. While the study is methodologically sound and clinically relevant, several areas require refinement to enhance clarity, validity, and applicability.

Strengths

1. Relevance and Novelty

o The topic addresses an important clinical dilemma: balancing the benefits of temporary pacing with the risks of unnecessary pacing wire insertion.

o The study attempts to fill a gap in current literature by providing a quantitative assessment model.

2. Methodological Rigor

o The study includes a large sample size (n=2326), enhancing statistical power.

o The use of MLR is appropriate for identifying independent risk factors.

o Sensitivity analysis and subgroup analysis strengthen the robustness of findings.

3. Development of a Practical Tool

o The scoring system offers a structured approach for identifying patients at low risk of requiring pacing, potentially reducing unnecessary interventions.

o The chosen cutoff (score ≤1) minimizes false negatives, ensuring that high-risk patients are not overlooked.

Areas for Improvement

1. Clarity in Presentation

o The manuscript presents results in a highly technical manner. Simplifying the text and using clearer tables or graphical representations (such as summary tables of key findings) would improve readability.

o The replacement of Table 2 with forest plots is helpful, but further effort is needed to make the statistical findings more accessible to clinicians.

2. Validation of the Scoring System

o The absence of external validation is a significant limitation. While the authors acknowledge this, they should provide a more detailed discussion on the necessity of validating the model in independent cohorts.

o The current validation approach (testing the scoring system within the same dataset) may introduce bias and overestimate model performance.

3. Consideration of Confounding Factors

o The study does not fully explore perioperative factors (e.g., medication use, intraoperative hemodynamics, and surgeon expertise) that may influence pacing needs.

o A more detailed discussion on how the type of surgery affects pacing risk is necessary. For example, why do non-CABG procedures increase the likelihood of pacing?

4. Limitations and Bias

o The manuscript acknowledges selection bias but does not sufficiently elaborate on recall bias, missing data issues, or potential inaccuracies in data collection.

o The retrospective design limits causal inference, which should be explicitly stated in relation to future prospective studies.

5. Clinical Implementation

o The discussion should elaborate on how the scoring system could be integrated into clinical workflows.

o Exploring the potential for incorporation into electronic health records (EHRs) for automated decision support would enhance practical utility.

Conclusion

This study contributes valuable insights into the risk assessment of temporary pacing post-CPB-assisted cardiovascular surgery. While the methodology is sound and the scoring system has potential clinical utility, improvements in clarity, external validation, and discussion of confounding factors would further strengthen its impact. Addressing these aspects would make the research more applicable to a broader cardiovascular surgical audience.

Reviewer #5: (No Response)

7. PLOS authors have the option to publish the peer review history of their article (what does this mean? ). If published, this will include your full peer review and any attached files.

**Do you want your identity to be public for this peer review?** For information about this choice, including consent withdrawal, please see our Privacy Policy .

Reviewer #2: **Yes: ** Victory Ashonibare

Reviewer #4: **Yes: ** Juliana Aggrey

Reviewer #5: No

---

## [Author Response · Author response to Decision Letter 2]

10 Apr 2025

Responses to the reviewers

(1)Reviewer #4

1. Clarity in Presentation

Q: The manuscript presents results in a highly technical manner. Simplifying the text and using clearer tables or graphical representations (such as summary tables of key findings) would improve readability. The replacement of Table 2 with forest plots is helpful, but further effort is needed to make the statistical findings more accessible to clinicians.

A: Thanks for your valuable advice over this time. We see that Plos ONE has a broad readership and added the schematics to illustrate the workflow of our study. Please see the new Fig 6 in this version.

2. Validation of the Scoring System

Q: The absence of external validation is a significant limitation. While the authors acknowledge this, they should provide a more detailed discussion on the necessity of validating the model in independent cohorts. The current validation approach (testing the scoring system within the same dataset) may introduce bias and overestimate model performance.

A: To further discuss the validation shortness in our study, we added these sentences in the limitation part “This is a single-center study, and the conclusions should be validated using external datasets. To avoid selection bias from the small positive case sample size, we did not separate a validation group. However, testing the scoring system within the same dataset may overestimate the model performance. Future studies on machine learning models for the universal predicting purpose should ensure a sufficient sample size needed for complete model evaluation.”.

3. Consideration of Confounding Factors

Q: The study does not fully explore perioperative factors (e.g., medication use, intraoperative hemodynamics, and surgeon expertise) that may influence pacing needs. A more detailed discussion on how the type of surgery affects pacing risk is necessary. For example, why do non-CABG procedures increase the likelihood of pacing?

A: To better explain the increased risk of pacing in non-CABG procedures, which are in fact the valvular surgery here, we rewrote the Discussion 4.3 part as following,

“Valvular surgery is acknowledged to be a risk factor of AV block and can increase the risk of temporary pacing. From an anatomy perspective, valve-related procedures often carry a risk of AV block due to its proximity of the AV conduction axis, and surgical operations can cause iatrogenic injuries[21]. Technical advancements are needed to prevent such injuries. In contrary, CABG only has operations on the large branches of coronary arteries on the surface of the heart, which hardly causes damages to the conduction system in the deep layer, less requiring pacing compared with the valvular surgery. The risk factors of AV block after CABG includes old age, CPB factors and medication[22]. The future studies may group operation types into several categories according to these features or develop specialized evaluations for high risk procedures.”,

We believe that these sentences can explain the reason through the convincing anatomical aspects.

Additionally, the anti-atrial fibrillation medication is an important confounding factor, so we added this sentence in the corresponding part

“Additionally, the influence of anti-atrial fibrillation medication should be considered as a confounding factor.”

However, considering (1) other researches also identified that atrial fibrillation can cause postoperative cardiac arrest or similar events, and (2) the mechanisms explaining why atrial fibrillation increases the events’ risk are reasonable, we do not conclude that the medication confounding can overturn our results. This is not the same case as we have between atrial fibrillation and its ablation.

The intraoperative hemodynamics can be quite vague in our topic. In the CPB assisted operations, the blood flow is in a special circulation, and we believe that the factor inclusion of CPB pump, circulation type, hypothermia, cardioplegia (volume and type) may be sufficient to indicate its statues. As far as we know, there is no study including these factors on temporary pacing. We agree a better intraoperative hemodynamic assessment is necessary, as we did not find independent risk factors within the selected parameters, while another study reported that pulmonary hypertension was a risk factor.

4. Limitations and Bias

Q: The manuscript acknowledges selection bias but does not sufficiently elaborate on recall bias, missing data issues, or potential inaccuracies in data collection. The retrospective design limits causal inference, which should be explicitly stated in relation to future prospective studies.

A: Although this is a retrospective study, our data were specially collected in the acknowledged clinical projects guided by corresponding methods, and therefore we believe that we minimized the recall bias, which has been emphasized in the Method 2.1 part in this revision. Usually, the missing data issues and inaccuracies in data collection can be concluded as the so called “information bias”, and they were stated in the Discussion 4.5 part with the specific sentence “information (including missing data bias caused by the update of recording systems, ascertainment bias in diagnoses, and personal factors)”. In this revision we also added the personal factors that were obviously not avoidable here. We believe that the inaccuracies in data collection were mostly the ascertainment bias in diagnoses, because although the indication of pacing was clear, it is hard to ensure fully unified diagnoses from different persons according to the ECG and cardiac output results. In the Discussion 4.5 part, we advocate having multi-center cohort studies on currently consensus factors for better causal inference, and we believe that with the new technologies developed (e.g. the millimetre-scale bioresorbable optoelectronic system for temporary pacing reported recently in Nature), this issue may be well investigated with minimized selection bias through RCT designs.

5. Clinical Implementation

Q: The discussion should elaborate on how the scoring system could be integrated into clinical workflows. Exploring the potential for incorporation into electronic health records (EHRs) for automated decision support would enhance practical utility.

A: We added a Discussion 4.4 part to discuss the clinical implementation of our study. We found your suggestion of incorporation of EHRs enlightening, and therefore we have this sentence at the end of this part “The ideal evaluation of pacing need should be simple, effective, clinically appropriate, and incorporating local epidemiological data into electronic health records to enable automated decision support would significantly enhance clinical utility”.

---

## [Editor Report · Decision Letter 2]

15 Apr 2025

Risk assessment of temporary pacing for cardiac arrest after cardiopulmonary bypass-assisted cardiovascular surgery: a case-control study

PONE-D-24-58174R2

Dear Dr. Wang,

We’re pleased to inform you that your manuscript has been judged scientifically suitable for publication and will be formally accepted for publication once it meets all outstanding technical requirements.

Kind regards,

Eyüp Serhat Çalık

Academic Editor

PLOS ONE
---

## [Editor Report · Acceptance letter]

PONE-D-24-58174R2

PLOS ONE

Dear Dr. Wang,

I'm pleased to inform you that your manuscript has been deemed suitable for publication in PLOS ONE. Congratulations! Your manuscript is now being handed over to our production team.

Kind regards,

on behalf of

Dr. Eyüp Serhat Çalık

Academic Editor

PLOS ONE